# Evaluation of the Quality of Information Available on the Internet Regarding Chronic Ankle Instability

**DOI:** 10.3390/medicina58101315

**Published:** 2022-09-20

**Authors:** Sung-Joon Yoon, Jun-Bum Kim, Ki-Jin Jung, Hee-Jun Chang, Yong-Cheol Hong, Chang-Hwa Hong, Byung-Ryul Lee, Eui-Dong Yeo, Hong-Seop Lee, Sung-Hun Won, Jae-Young Ji, Dhong-Won Lee, Woo-Jong Kim

**Affiliations:** 1Department of Orthopaedic Surgery, Soonchunhyang University Hospital Cheonan, 31 Suncheonhyang 6-gil, Dongam-gu, Cheonan 31151, Korea; 2Department of Orthopaedic Surgery, Veterans Health Service Medical Center, Seoul 05368, Korea; 3Department of Foot and Ankle Surgery, Nowon Eulji Medical Center, Eulji University, 68 Hangeulbiseok-ro, Nowon-gu, Seoul 01830, Korea; 4Department of Orthopaedic Surgery, Soonchunhyang University Hospital Seoul, 59 Daesagwan-ro, Yongsan-gu, Seoul 04401, Korea; 5Department of Anesthesiology and Pain Medicine, Soonchunhyang University Hospital Cheonan, 31 Suncheonhyang 6-gil, Dongam-gu, Cheonan 31151, Korea; 6Department of Orthopaedic Surgery, Konkuk University Medical Center, 120-1 Neungdong-ro, Gwangjin-gu, Seoul 05030, Korea

**Keywords:** chronic ankle instability, Internet medical information, website assessment

## Abstract

*Background and objectives*: Most Koreans obtain medical information from the Internet. Despite the vast amount of information available, there is a possibility that patients acquire false information or are dissatisfied. Chronic ankle instability (CAI) is one of the most common sports injuries that develops after an ankle sprain. Although the information available on the Internet related to CAI has been evaluated in other countries, such studies have not been conducted in Korea. *Materials and Methods*: The key term “chronic ankle instability” was searched on the three most commonly used search engines in Korea. The top 150 website results were classified into university hospital, private hospital, commercial, non-commercial, and unspecified websites by a single investigator. The websites were rated according to the quality of information using the DISCERN instrument, accuracy score, and exhaustivity score. *Results*: Of the 150 websites, 96 were included in the analysis. University and private hospital websites had significantly higher DISCERN, accuracy, and exhaustivity scores compared to the other websites. *Conclusions*: Accurate medical information is essential for improving patient satisfaction and treatment outcomes. The quality of websites should be improved to provide high-quality medical information to patients, which can be facilitated by doctors.

## 1. Introduction

According to the 2018 report by the International Telecommunication Union, the Internet usage rate in Korea is 95.1%, which is the highest in the world, and most people can use the Internet easily [1]. In addition, patients frequently use the Internet to access medical information. A study found that almost 90% and 65% of patients search the Internet to obtain information regarding their disease and its treatment [2]. Patients who obtain accurate medical information through the Internet tend to enter into good patient–physician relationships, which leads to better treatment outcomes [3,4]. Therefore, qualitative analyses of the quality of such websites are necessary to ensure that they provide accurate medical information to patients.

Chronic ankle instability (CAI) is one of the most common sports injuries that develops after an ankle sprain [5]. Ankle sprains are the most common musculoskeletal sports injuries, with a prevalence of about 25,000 to 30,000 people per day in the United States, accounting for about 15~25% of all sports injuries. There are no accurate statistics on ankle sprains in Korea, but it can be assumed that they are not much different from other countries. With acute lateral ankle sprains, good results can be obtained through conservative treatment in most cases, but it is known that some patients will develop CAI, i.e., re-injury or persistent symptoms resulting therefrom. Diminished range of motion (ROM), decreased strength, altered functional movement patterns, and impaired neuromuscular control are characteristics of CAI [6]. A decreased ankle dorsiflexion ROM increases instability, which directly cause the loss balance [7]. Additionally, in the presence of ankle instability, the proximal muscles of the lower limb may be affected [8]. A recent review found that 70% of individuals who experience an acute lateral ankle sprain may develop CAI over a short time period, and a prospective cohort study found that 40% of individuals develop CAI 1 year after a first-time lateral ankle sprain [9,10,11]. In CAI, unlike acute ankle sprain, if conservative treatment is ineffective, surgical treatment should be considered [6,12,13,14,15].

Since CAI is an important pathological condition that can cause discomfort and joint destruction in the long term, accurate information and understanding are essential for proper diagnosis and treatment [16]. Although the information available on the Internet related to CAI has been evaluated in other countries, such studies have not been conducted in Korea [17]. Therefore, we hypothesized that the quality of information would vary among websites and assessed the quality of Internet websites that provide information related to CAI.

## 2. Materials and Methods

### 2.1. Study Subjects and Criteria

This study was conducted at Soonchunhyang University Hospital, Cheonan, South Korea. According to Internet Trend (www.internettrend.co.kr (accessed on 29 August 2022)), a Korean log analysis website, the top three search engines (Naver, Google, and Daum) accounted for 56.10%, 34.73%, and 5.46% of the search traffic in 2021, respectively; the three search engines combined accounted for 96.29% of search traffic. Hence, these three search engines were searched for “chronic ankle instability” and the top 50 websites from each search engine were selected for analysis. In all, 150 websites were analyzed. Website contents that were not in text format, such as radio or TV broadcast materials and YouTube, were excluded from the study. Websites that could not be accessed, duplicate websites, and websites not related to CAI were also excluded. Figure 1 shows the method of selection and evaluation of websites.

### 2.2. Website Classification

Website authors were classified into university hospitals, private hospitals, commercial organizations (such as medical device company), non-commercial organizations (such as civic groups and government agencies), and others. In addition, they were classified by specialty: orthopedic surgeons, rehabilitation medicine specialists, other doctors, oriental doctors, and non-doctor authors.

### 2.3. Evaluation of Website Information

Although several studies have evaluated the medical information available on websites, there is no standard evaluation tool. In the present study, we used the DISCERN instrument, which is commonly used for the objective evaluation of websites (Table 1) [18]. DISCERN is a tool developed by Oxford University and British Library for qualitative evaluation of medical information and assigns a score of 1–5 for 16 items (total score: 16–80) [19,20,21,22,23,24,25,26]. We modified it to a score of 20. Because DISCERN does not evaluate the accuracy and exhaustivity of information, we used five additional items to evaluate these factors (Table 2 and Table 3). Two orthopedic surgeons with training in podiatry and two sports rehabilitation medicine specialists independently evaluated the websites and assigned the scores.

### 2.4. Statistical Analysis

Statistical analysis was performed using SPSS (version 26.0; IBM Corp., Armonk, NY, USA). One-way analysis of variance and the Kruskal–Wallis test were performed, as appropriate. The significance level was set at *p* < 0.05.

## 3. Results

### 3.1. Website Classification

Of the 150 websites identified, 96 were included in the analysis. In total, 19.7%, 32.2%, 18.7%, 20.8%, and 8.3% were published by university hospitals, private hospitals, commercial organizations, non-commercial organizations, and other organizations, respectively (Figure 2). In addition, 28.1%, 21.8%, 20.8%, 16.6%, and 12.5% of the authors were orthopedic surgeons, rehabilitation medicine specialists, other doctors, oriental doctors, and non-doctors, respectively (Figure 3). University and private hospitals accounted over half of the authors (51.9%). Orthopedic surgeons and rehabilitation medicine specialists accounted for nearly half of the authors (49.9%). The combination these of these data is summarized in Table 4.

### 3.2. Evaluation of Website Information

The mean DISCERN, accuracy, and exhaustivity scores assigned by the four evaluators were recorded. The website scores were compared to the mean score of the group to which the website was assigned. The converted DISCERN scores were 15.4, 14.6, 9.4, 8.7, and 6.3 for websites published by university hospitals, private hospitals, non-commercial organizations, commercial organizations, and others (Table 5). The accuracy scores were 4.6, 4.3, 3.8, 3.6, and 2.8 for websites published by university hospitals, private hospitals, non-commercial organizations, commercial organizations, and others (Table 6). The exhaustivity scores were 4.4, 4.1, 3.6, 3.4, and 2.1 for websites published by university hospitals, private hospitals, non-commercial organizations, commercial organizations, and others (Table 7). The DISCERN scores of the university and private hospital websites were significantly higher than those of the other three groups (non-commercial organization, commercial, and other sites) (*p* = 0.04); there were no statistically significant differences between the scores of university and private hospital websites (*p* = 0.21). In addition, the accuracy scores of university and private hospital sites were significantly higher than those of the other three groups (*p* = 0.03); there were no statistically significant differences between the accuracy scores of university and private hospital websites (*p* = 0.33). The exhaustivity scores of the university and private hospital websites were significantly higher than those of the other groups (*p* = 0.04); there were no statistically significant differences between the exhaustivity scores of university and private hospital websites (*p* = 0.34).

## 4. Discussion

Several studies have reported that patient education and provision of accurate information improve treatment outcomes [27,28,29]. The Internet represents a cost-effective medium through which patients can be educated. Vast amounts of information are available on the Internet but it is essential to select accurate and reliable information [30,31,32]. In such an environment, the provision of correct medical information to patients by medical professionals is essential for a desirable patient–physician relationship, and plays a decisive role in improving patient satisfaction and treatment compliance [33]. However, unlike peer-reviewed academic journals, anyone can post information on websites, which can lead to patients obtaining low-quality medical information, poor decision making, and even treatment failure [25,34]. Because of the frequent use of Internet websites for patient education, several studies have evaluated the medical information available on the Internet; however, such studies have rarely been performed in Korea.

CAI is a major complication of acute ankle sprain that leads to discomfort during daily and sports activities [35]. Studies of online information related to lumbar disc herniation, lumbar stenosis, and cervical disc herniation have been published in the field of orthopedic surgery in Korea, but no previous study from Korea has evaluated the information related to CAI available on the Internet [16,18,36,37]. In addition, some of the aforementioned domestic studies conducted online information quality analyses using evaluation criteria arbitrarily created by the authors; thus, objectivity was not guaranteed and the criteria were applied only once. As a result, no follow-up studies were conducted and there is little possibility of revising or developing the evaluation criteria. By contrast, in this study, reliability and objectivity could be ensured by using the internationally standardized DISCERN instrument, which has also been used in other medical papers. Thus, this study should lead to follow-up research, rather than acting as a standalone study.

This study created accuracy and exhaustivity items; similar items have been constructed and analyzed in other studies to supplement DISCERN. It seems necessary to improve the DISCERN instrument [38,39]. As DISCERN is used as a standard to evaluate medical text, we excluded audiovisual materials, such as radio/TV broadcasts and YouTube videos; i.e., we used the DISCERN instrument as the sole evaluation standard [40]. In subsequent studies, standards that overcome the limitations of the DISCERN instrument, and the use of both audiovisual and written data will be needed [41]. University and private hospital websites had higher DISCERN, accuracy, and exhaustivity scores compared to other websites; these results are in line with previous studies [42,43]. However, some studies have suggested that academic institution websites, such as university hospital websites, may not necessarily provide high-quality information. Although it is extremely rare for an academic institution to provide inaccurate information, there is a tendency to omit information related to treatments that are not offered at the institution [44,45].

In this study, duplicate websites identified using the top three search engines were excluded. Comparing the order of presentation of websites within the search results for each search engine, and the total DISCERN, accuracy, and exhaustivity scores, would enable the evaluation of the ability of search engines to list websites that provide high-quality information first.

We analyzed the quality, accuracy, and exhaustivity of the information provided by websites to determine the quality of the information provided to patients. However, we did not evaluate the comprehensibility and readability of information, which are important for patient education. Information must not only be reliable; it must be understandable to patients [46]. We did not evaluate whether medical consumers with poor health literacy could easily understand the information, which is necessary for patient education. Future studies should evaluate comprehensibility and readability to allow a comprehensive evaluation of medical information for patient education. In the present study, the information provided by the websites was analyzed by two orthopedic surgeons trained in podiatry and two sports rehabilitation medicine specialists. A limitation of the present study is that the information was not evaluated by doctors from other departments; this should be addressed in future studies.

Many studies used qualitative evaluation methods and have developed certification systems for use in Korea to ensure the quality of medical information is of high quality [47]. As a method to provide high-quality Internet medical information to medical consumers, medical organizations could implement a “quality labeling system”; a representative example of this is the Health On the Net (HON) Foundation code [48]. However, due to the lack of awareness of such certification systems, many high-quality websites are not certified [25,34]. Some studies have also shown that certification systems do not guarantee the quality of online medical information [43]. As the amount of Internet medical information increases, medical consumers are experiencing difficulty obtaining reliable information. There is a research result that about 70% of patients hope to be recommended an appropriate medical sites to their doctors; medical organizations should play a central role in ensuring high-quality medical information [31].

Finally, detection bias in association with evaluations of individual websites should be reduced in future studies.

## 5. Conclusions

Accurate medical information should be obtained from the Internet to improve patient–physician relationships and treatment outcomes. However, regarding CAI, there is a lack of research on the quality of such information in Korea. Information provided by university and private hospital websites had the highest accuracy and quality scores. Efforts should be made to evaluate the quality of medical information provided by websites. It is expected to contribute to improving the quality of information regarding CAI on the Internet. Medical organizations and physicians should play a central role in improving the information available to patients.

## Figures and Tables

**Figure 1 medicina-58-01315-f001:**
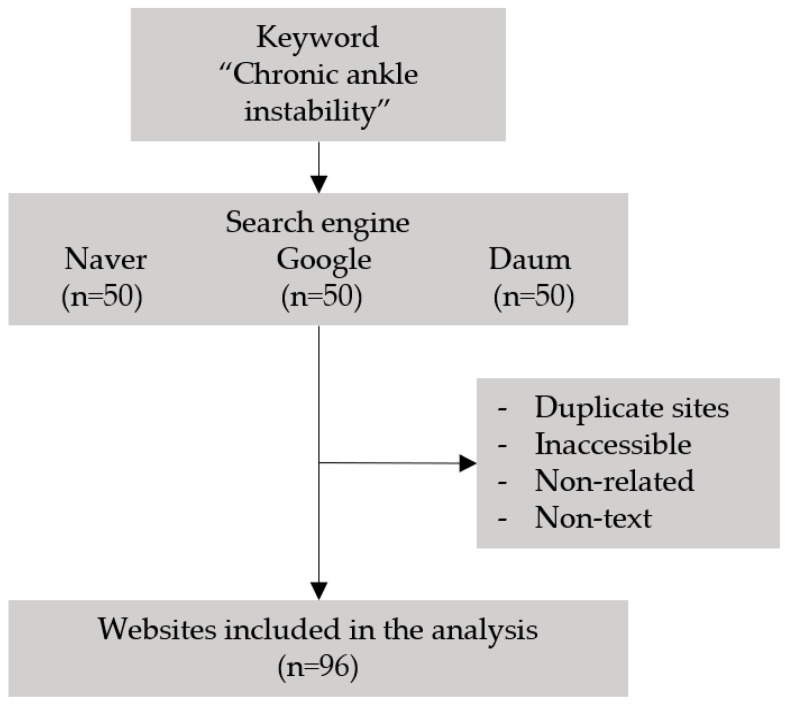
Method of selection and evaluation of websites.

**Figure 2 medicina-58-01315-f002:**
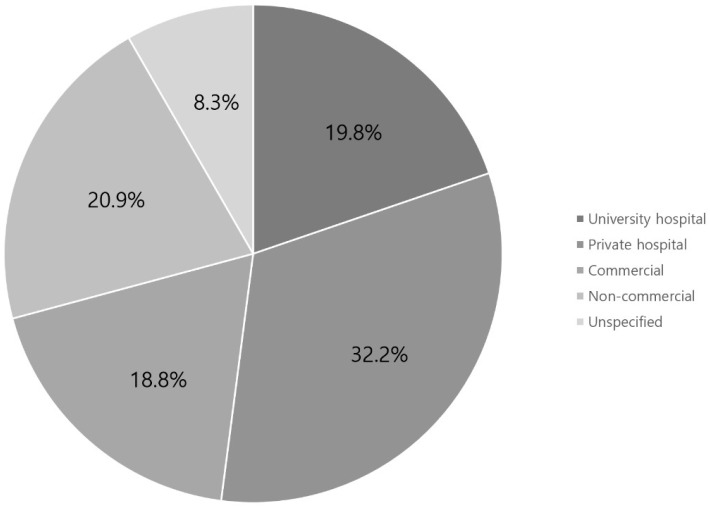
Summary of websites by website authors.

**Figure 3 medicina-58-01315-f003:**
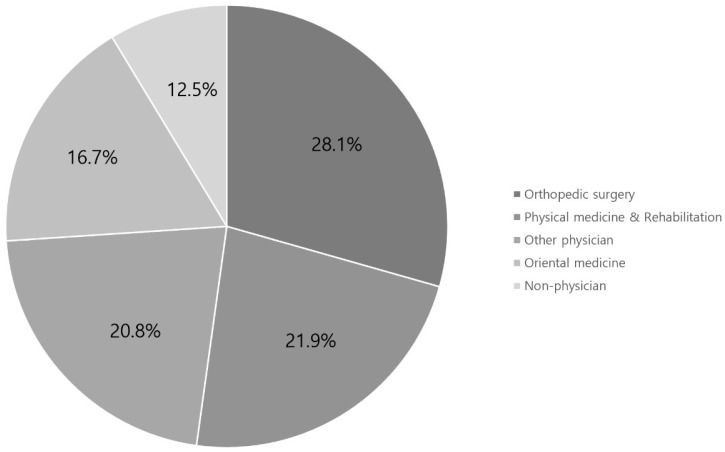
Summary of websites by specialty.

**Table 1 medicina-58-01315-t001:** DISCERN questionnaire for the evaluation of consumer health information.

Question	Score
1. Are the aims clear?	1–5 points
2. Does it achieve its aims?	1–5 points
3. Is it relevant?	1–5 points
4. Is it clear what sources of information were used to compile the publication (other than the author or producer)?	1–5 points
5. Is it clear when the information used or reported in the publication was produced?	1–5 points
6. Is it balanced and unbiased?	1–5 points
7. Does it provide details of additional sources of support and information?	1–5 points
8. Does it refer to areas of uncertainty?	1–5 points
9. Does it describe how each treatment works?	1–5 points
10. Does it describe the benefits of each treatment?	1–5 points
11. Does it describe the risks of each treatment?	1–5 points
12. Does it describe what would happen if no treatment was used?	1–5 points
13. Does it describe how the treatment choices affect overall quality of life?	1–5 points
14. Is it clear that there may be more than one possible treatment choice?	1–5 points
15. Does it provide support for shared decision making?	1–5 points
16. Based on the answers to all of the above questions, rate the overall quality of the publication as a source of information about treatment choices	1–5 points
Total	16–80 points

**Table 2 medicina-58-01315-t002:** Accuracy score scale.

No inaccurate information	5 points
Rare inaccurate information without consequences	4 points
Frequent inaccurate information without consequences	3 points
One serious inaccuracy	2 points
Many serious inaccuracies	1 point

**Table 3 medicina-58-01315-t003:** Exhaustivity score scale.

Exhaustive subject coverage	5 points
Complete subject coverage but with few details	4 points
Partial coverage of subject with sufficient details	3 points
Partial coverage of subject with few details	2 points
Little coverage of the subject matter	1 point

**Table 4 medicina-58-01315-t004:** Websites classified by authorship and specialty.

	Orthopedic Surgery	Physical Medicineand Rehabilitation	Other Physicians	Oriental Medicine	Non-Physician	Total
University hospital	16	3	-	-	-	19
Private hospital	8	12	4	7	-	31
Commercial	1	2	2	6	7	18
Non-commercial	2	3	9	1	5	20
Others	-	1	5	2	-	8
Total	27	21	20	16	12	-

**Table 5 medicina-58-01315-t005:** DISCERN scores by website authors.

Website Authorship	DISCERN Score ^1^
University hospital ^2,3^	15.4 (0.77)
Private hospital ^2,3^	14.6 (2.58)
Commercial ^2^	9.4 (0.91)
Non-commercial ^2^	8.7 (1.83)
Unspecified ^2^	6.3 (0.86)

^1^ Values are mean (SD). ^2^ University and private hospital websites scored significantly higher than those of the other three groups (*p* = 0.04). ^3^ There were no statistically significant differences between the scores of university and private hospital websites (*p* = 0.21).

**Table 6 medicina-58-01315-t006:** Accuracy scores by website authors.

Website Authorship	Accuracy Score ^1^
University hospital ^2,3^	4.6 (0.49)
Private hospital ^2,3^	4.3 (0.62)
Commercial ^2^	3.8 (0.87)
Non-commercial ^2^	3.6 (0.72)
Unspecified ^2^	2.8 (0.68)

^1^ Values are mean (SD). ^2^ University and private hospital websites scored significantly higher than those of the other three groups (*p* = 0.03). ^3^ There were no statistically significant differences between the scores of university and private hospital websites (*p* = 0.33).

**Table 7 medicina-58-01315-t007:** Exhaustivity scores by website authors.

Website Authorship	Exhaustivity Score ^1^
University hospital ^2,3^	4.4 (0.63)
Private hospital ^2,3^	4.1 (0.76)
Commercial ^2^	3.6 (0.59)
Non-commercial ^2^	3.4 (0.86)
Unspecified ^2^	2.1 (0.92)

^1^ Values are mean (SD). ^2^ University and private hospital websites scored significantly higher than those of the other three groups (*p* = 0.04). ^3^ There were no statistically significant differences between the scores of university and private hospital websites (*p* = 0.34).

## Data Availability

Data sharing is not applicable to this article as no datasets were generated or analyzed during the current study.

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
