# Peer review of "Evaluation of the Quality of Information Available on the Internet Regarding Chronic Ankle Instability"

_medicina, 2022, doi:10.3390/medicina58101315_

Round 1

Reviewer 1 Report

Thank you very much for this very interesting work with highly innovative methods.

I have found your article particularly well written.

I have only one question which response may help readers. Indeed, you have classified the web-sites according to their authors and according to specialty.

I would like to know the results of the combination of these 2 data. For example, who are the writers of the commercial websites, of the non-commercials ones, the academic ones, etc.

It would give interesting information.

Reviewer 2 Report

The topic is one of importance given the high number of presentations to health services that are related to concerns on   Chronic Ankle Instability. Also, this is an interesting aim for analyse a quality assessment with a focus on websites that provide information regarding CAI and its treatment options. I think it would be a more clear study if the following parts were revised and supplemented. These will be discussed below relative to the information of the manuscript.

General Comments:

Overall the manuscript is generally well written and is a topic of interest. However after reading it a number of times I am still left without key take-home points. I believe these points are in the results they just need to be developed.

Specific comments:

Abstract:

1) The authors state few studies have evaluated the quality of the information available on the Internet regarding chronic ankle instability  and they conducted a quality assessment with a focus on websites that provide information regarding CAI and its treatment options. This seems like too much of an over simplification of what was done. I do feel that it would be beneficial to explain what specifically you are looking at in relation to chronic ankle instability (this also applies to the main body of the paper). Is it the development of chronic ankle instability condition  literature. This needs to be made clearer throughout the paper. (Major Compulsory Revision)

Introduction

2) The first paragraph should have a sentence or two added that better outlines why this study is important related with activity patterns during external perturbation in subjects with and without functional ankle instability https://pubmed.ncbi.nlm.nih.gov/28843163/ (Major Compulsory Revision)

The authors do a poor job on reviewing relevant literatura related with importance with  ankle dorsiflexion range of motion. Please revise the research of Romero  et al https://pubmed.ncbi.nlm.nih.gov/28070457/

3) In the last paragraph, the significance of the proposed word should be included highlighting why your work is important. what is the scientific contribution of this paper? it is not clear how this paper can make a significant contribution to the state of the art. (Major Compulsory Revision).

In addition, author´s hypotheses should be included.

5) This methods section is poor, needs to present a better rationale for the study and the methodology employed. Also, neither appear information related with inclusion and exclusion criteria, dates, protocol. The study design is a systematic review, where the study guidelines? This research adhere to reporting Prisma guidelines? (Major Compulsory Revision).

6) Where the experiments carried out? In a hospital? In an educational institution? Provide this information.

7) Add a study flow chart for the readers. (Major Compulsory Revision).

8) Include p-values in all the tables (Major Compulsory Revision).

9) Please reconsider listing of the all figures and remove this figures are not informative and include this info as Tables with p values.

10) The Discussion section is a rehashing of the results. It does not appear that the authors include much interpretation of what the study findings mean for clinical practice or research. (Major Compulsory Revision)

FInally, the conclusión is weak and too long. (Major Compulsory Revision)

Round 2

Reviewer 2 Report

I thank the corresponding author for their comments. I have read through the subsequent changes made to the manuscript and I have no further comments or suggestions.